# Current Understanding of Role of Vesicular Transport in Salt Secretion by Salt Glands in Recretohalophytes

**DOI:** 10.3390/ijms22042203

**Published:** 2021-02-23

**Authors:** Chaoxia Lu, Fang Yuan, Jianrong Guo, Guoliang Han, Chengfeng Wang, Min Chen, Baoshan Wang

**Affiliations:** Shandong Provincial Key Laboratory of Plant Stress Research, College of Life Sciences, Shandong Normal University, Jinan 250014, China; yingks_1982@126.com (C.L.); yuanfang@sdnu.edu.cn (F.Y.); gjr20022008@163.com (J.G.); gl_han@163.com (G.H.); wywcf123@163.com (C.W.); chenminrundong@126.com (M.C.)

**Keywords:** recretohalophytes, salt resistance, salt gland, salt secretion, vesicular trafficking, vesicle

## Abstract

Soil salinization is a serious and growing problem around the world. Some plants, recognized as the recretohalophytes, can normally grow on saline–alkali soil without adverse effects by secreting excessive salt out of the body. The elucidation of the salt secretion process is of great significance for understanding the salt tolerance mechanism adopted by the recretohalophytes. Between the 1950s and the 1970s, three hypotheses, including the osmotic potential hypothesis, the transfer system similar to liquid flow in animals, and vesicle-mediated exocytosis, were proposed to explain the salt secretion process of plant salt glands. More recently, increasing evidence has indicated that vesicular transport plays vital roles in salt secretion of recretohalophytes. Here, we summarize recent findings, especially regarding the molecular evidence on the functional roles of vesicular trafficking in the salt secretion process of plant salt glands. A model of salt secretion in salt gland is also proposed.

## 1. Introduction

Soil salinization is increasingly becoming a serious problem as well as a major constraint for global agricultural production. At present, about one-fifth of the world’s irrigated land, approximately 950 million hectares, is affected by salinization [1,2]. In recent years, the secondary salinization of soil has increased due to the poor management of soil and water resources, overgrazing, intensive agriculture, industrial pollution, climate change, and so on [3]. On one hand, the world’s population has increased rapidly. On the other hand, arable land has gradually decreased due to climate change, soil salinization, and industrialization. Therefore, it is a big challenge to meet the needs of the food increase worldwide. Unraveling the salt resistance mechanisms of the halophytes to provide resources for the breeding of salt-tolerant crops is therefore one of the key strategies for the utilization and sustainable development of saline–alkali land.

The main characteristic of saline–alkali land is the high content of salt ions (mainly Na^+^ and Cl^−^), which interferes with metabolic processes and inhibits growth and development of crops, leading to reduction of yield, and even resulting in crop death in severe cases [4,5]. Most terrestrial plants are sensitive to salt, while just 1% of plant species can overcome osmotic and ion stresses to grow and develop normally in saline environments. Plants surviving in salt concentrations of 200 mM and above are called halophytes, and can accumulate salt concentrations of 500 mM or more in their leaves and shoots [6,7]. All other plants are salt sensitive, and referred to as nonhalophytes (formerly glycophytes). The enzymes in the cytosol of the halophytes are as sensitive to salt as those of the nonhalophytes, and they are simply not exposed to concentrations of salt higher than 100–200 mM in the cytoplasm of halophytes [8,9]. To avoid the toxicity of high salinity on their metabolisms, halophytes have evolved a series of salt tolerance mechanisms to sequester excessive salt into the vacuole or secrete them out. In the last three decades, much progress has been made in understanding the molecular mechanisms involved in the compartmentalization of excessive salt ions, such as the movement of Na^+^ into the vacuoles or out of the cells via the salt-overly-sensitive (SOS) pathway [10,11,12]. However, little is known about the molecular mechanisms of salt secretion.

Halophytes can belong to three categories: (1) Euhalophytes, which dilute any absorbed salt in their succulent leaves [13]; (2) salt excluders, which avoid the uptake of salt or shed leaves containing toxic levels of salt [14]; (3) recretohalophytes, which possess unique salt-secreting structures [15,16,17]. Simply stated, there can be two types of secreting structures: The exo-recretohalophytes (e.g., *Limonium bicolor*) possessing salt glands, and the endo-recretohalophytes (e.g., *Mesembryanthemum crystallinum*) with salt bladders [3,18]. Scattered across the surfaces of recretohalophyte leaves and stems, the salt glands and salt bladders are highly evolved structures with the main function of secreting excessive salt outside the body to avoid high levels of accumulation in the cells, enabling the plants to survive on saline–alkali soil without being damaged [6,19].

Massive work has been done to elucidate the mechanisms of salt gland secretion. Three hypotheses were historically suggested to explain the mechanism of recretohalophyte salt secretion on the basis of histochemistry and ultrastructure: (1) Arisz et al. (1955) proposed a role for osmotic potential in the salt secretion of *Limonium latifolium* [20]; (2) Levering and Thomson (1971) suggested that a transfer system similar to liquid flow in animals was involved in salt secretion in *Spartina foliosa* [21]; and (3) Ziegler and Lüttge (1967) proposed that the salt solution was secreted via vesicle-mediated exocytosis [22,23]. The first hypothesis proposed that the active accumulation of ions in the salt glands resulted in a decrease in the osmotic pressure of the gland cells, leading to a significant increase in hydrostatic pressure. When the pressure in the salt gland reaches the highest level, the gland cells periodically form droplets through the secretory pores to relieve the hydrostatic pressure, thereby expelling ions out of the salt gland [3,24]. The ultrastructure of *Distichlis spicata* supports this hypothesis, as its salt glands possess only a single, thin membrane that resembles a valve separating the parietal layer of the cytoplasm [25]. In terms of the second hypothesis, researchers supposed that an extensive membrane system is formed in the basal cell of *Spartina anglica*, and ions are secreted out of salt glands through ion channels on the membrane, similar to the absorption and secretion of animal tissue epithelial cells. Some studies showed that salt secretion was significantly reduced in *Tamarix* species and *Chloris gayana* when treated with ouabain [26], a specific Na^+^-ATPase inhibitor in animal that enhances K^+^ influx and inhibits Na^+^ efflux by interacting with external K^+^ binding sites [27]. However, no genes encoding plasma membrane-bound Na^+^-ATPases have yet been identified in genome-sequenced higher plants. The third hypothesis, in which vesicles are involved in the secretion of salt from the salt glands, is supported by observations of the *Limonium bicolor* and *Tamarix usneoides* salt glands made using scanning electron microscopy (SEM) [28,29]. These studies showed electron-dense materials primarily accumulated in the vesicles, particularly in plants treated with salt, and a great deal of these vesicles fused with the plasma membrane.

Vesicular trafficking is the coordinated process of moving a membrane-enclosed substance in cell, which is conserved in the eukaryotes and plays indispensable roles in the function of living cells [30,31], including the development of plant, the plant immunity, plant cytokinesis, the elongation of pollen tubes, and the secretion of nectar, sugars, and phenols [32,33,34]. Furthermore, there is substantial evidence that vesicular transport plays a crucial role in salt secretion from plant salt glands (Figure 1). In the following paragraphs, the results of ultrastructural, histochemical, physiological, transcriptome and genomic sequencing, and molecular biology studies of salt secretion are discussed in detail.

## 2. Structural Features of Salt Glands

Salt glands and bladders are the unique morphological and structural characteristics of recretohalophytes [3], and are specialized epidermal cells, homologous to trichomes [35]. Salt glands directly secrete salt to the leaf surface, with a cuticle-lined structure, autofluorescence under UV excitation [36,37]. Salt bladders collect salt in a specialized vacuolated bladder cell, which may rupture in certain situations [38,39]. Salt glands have different structures in different species, and the number of cells that make up the salt gland is different. In general, the salt glands are multicellular in dicotyledonous recretohalophytes (collecting cells, outer cup cells, inner cup cells, accessory cells, and secretory cells), and bi-cellular in monocotyledonous recretohalophytes (basal cell and cap cell), while these salt glands have common characteristics [3,18,24].

Our recent understanding on the secretory mode of the salt glands is mainly based on observation of their ultrastructural differences during secretion. The cells of the secretory tissue are ordinarily characterized by the presence of many small vesicles, a large nucleus, a relatively dense cytoplasm containing many Golgi apparatus and mitochondria, and numerous plasmodesmata between cells [24,28,40]. As a great number of vesicles and small vacuoles occur in secretory cells, many of them are located in the peripheral cytoplasm, and several studies have proposed that these vesicles might be involved in the secretion process [41,42,43]. Shimony et al. (1973) suggested that vesicles and membrane bands originate from the Golgi apparatus, which was consistent with the lanthanum tracing findings of Smaoui et al. (2011). Lanthanum is a barrier to apoplastic transport by the in situ precipitation of La^3+^ [42,44,45].

Electron-microscopic imaging of *Tamarix aphylla* salt glands treated with high concentrations of Na^+^ and K^+^ ions showed that many of the microvacuoles were associated with wall protuberances or with the plasma membrane [41,46]. Researchers found that, during salt secretion, the ions were contained within small vesicles. These vesicles were tightly bound to the cell wall and fused with the plasma membrane, and then the released substance passed through the apical cell wall and the cuticle to the environment as small entities [3,28,47,48,49]. Texas Red^®^-X phalloidin, a fluorescent probe-targeting stabilized F-actin, was used to stain the salt glands of *Avicennia officinalis*, revealing that abundant actin-like filaments can be seen in the secretory cells, especially protruding from the edge of the secretory cells towards the periclinal walls close to the cuticle layer or along the periphery of the secretory cells [40]. This implied vesicular transport was very active in the secretory cells.

To explore the effect of salt on the structure of the salt gland, some researchers compared the differences in the structure and secretion level of the salt glands in plants under control and salt-treated conditions. Salt application was shown to increase the number of vesicles and density of other organelles [29,47,48,50,51]. For *Frankenia*, there was a marked disparity in the ultrastructure of the salt glands between the control plants and those treated with high concentration of salt, with the latter accumulating more vesicles in their salt glands than the controls [52]. Mitochondria are usually observed in the salt gland cells, and are regarded as a supplier of energy for the ion transport involved in the excretion process [48]. Compared with the control, the volume of the mitochondrial fraction was increased in *Limonium platyphyllum* and *Avicennia germinans* [48,51]. Wilson et al. (2017) observed sequential micrographs of *Tamarix usneoides* salt gland cells, revealing that ions were removed from plant tissues by “parceling” [29]. These results indicate that plant salt glands are adapted to receive, accumulate, and effectively exclude ions from the leaves.

## 3. Physiological Studies of Salt Glands

Salt glands and bladders play a pivotal role in maintaining the ion balance, regulating the stability of osmotic pressure, and improving the salt tolerance of plants [6,28,37,53]. Since the salt glands are mostly distributed near the veins of the leaves [54], the excess ions can reach them more efficiently when transported by transpiration, as there is a layer of cuticle around the salt glands, and ions in the leaves can only reach the salt glands through the symplastic transport pathway [3,22]. The ions are then excreted by the secretory pores via a complex transport system between the constituent cells in the salt glands. Secretion is thus an active process that protects plant tissues from toxic ions [28].

During the process of gland secretion, however, water is lost when the ions are secreted [37,55]. The secretion products of the salt glands contain various ions, such as Na^+^, K^+^, Ca^2+^, Mg^2+^, Cl^−^, NO_3_^−^, SO_4_^2−^, and PO_4_^3−^ [56], as well as the less common ions Cs^+^, Rb^+^, Br^−^, I^−^, which are taken up by plants and subsequently secreted by the glands [57]. Shimony and Fahn (1968) speculated that pectic material was found in the secretions from the glands of *Tamarix*, while Pollak and Waisel (1970) concluded there are many organic materials, including proteins and free amino acids, in the secretions of *Aeluropus* [24,58]. Plants with salt secretion ability are often used to remove heavy metals, such as Cd, Pb, Cu, Zn, Al, and Fe, from soils, reducing heavy metal pollution [29,59,60,61,62,63]. These observations indicated that the secretion of plant salt glands is nonselective, and that the secreted ions are dependent on the salt composition in the rhizosphere [41,64].

While the selectivity of ions by ion channels and transporters is dependent upon the structural properties of the ion channels and transporters, the selectivity of vesicular ion transport is determined by the local ion concentration near the forming vesicle and the electric charge of the vesicle membrane is also nonselective [7,65]. This further illustrates that vesicle transport plays a certain role in salt secretion in the salt glands.

## 4. Cytological Studies of Salt Glands

Over the past few decades, there has been increasing evidence that vesicular trafficking is involved in effective and complicated secretory systems to deal with abiotic stress [33].

With the development of science and technology, the relationship between the structure and function of the salt glands has attracted much attention. As early as 1969, Thomson et al. observed the glands of *Tamarix aphylla* treated with Rb_2_SO_4_ using electron micrographs. Authors found that electron-dense material accumulated in the vesicles of the secretory cells, and speculated that microvesicles moved and eventually fused with the plasma membrane, releasing salt into the walls of the secretory cells [41]. Treating samples containing Cl^−^ with soluble Ag salts results in AgCl precipitation in plants, and these electron-dense deposits of Cl^−^ have been reported in some recretohalophytes, including *Limonium vulgare*, *Frankenia pulverulenta*, and *Tamarix aphylla* [22,52,66]. Cl^−^ was localized in the walls and vesicles, and particularly in protrusions in the cuticular chamber in *Sporobolus virginicus* [43]. According to a lanthanum-tracer study in *Atriplex halimus*, lanthanum was deposited in endocytosis vesicles, the nuclear envelope, the rough endoplasmic reticulum (ER), and often the largest Golgi vesicles [44]. Lu et al. (2020) treated *Limonium bicolor* leaves with brefeldin A (BFA), a specific inhibitor of Golgi-mediated secretion, which substantially decreased salt secretion. This treatment left the Golgi apparatus deformed, and the trans-Golgi network (TGN) was lost to the cytoplasm and, eventually, to the BFA compartments [67]. These data indicate that the Golgi apparatus plays a pivotal role in the process of salt gland secretion and vesicular transport participates in salt secretion.

The movement of the excessive ions within the salt gland cells requires that the vesicles are processed by the ER and labeled by the Golgi [29]. The salts are further concentrated or moved into the cuticular chamber where they migrate to the surface of leaves by diffusion.

## 5. Insights from Omics Applications

A systematic study of the recretohalophytes will help us to understand the molecular mechanisms of salt-gland secretion. Methods such as genomic DNA sequencing, RNA sequencing, and proteomics are important tools for plant gene mining in these studies.

Many recretohalophyte sequences have been analyzed thus far. An RNA-sequencing library from *Limonium bicolor* treated with 200 mM NaCl was an important resource for identifying the genes involved in salt secretion [68]. This transcriptomic analysis revealed that some vesicle-related genes were differentially expressed and were verified via qRT-PCR in increasing salt concentrations and secretion mutants. Tan et al. (2015) analyzed the proteome of the salt gland-enriched tissue from *Avicennia officinalis* using a cellular component analysis [69]. Some differential proteins were found in the tonoplast, mitochondria, Golgi apparatus, or the ER; more than 5% of them were found to be localized in the Golgi, and GTP-binding proteins located in the Golgi and the proton pump located in the vesicles of NaCl-treated plants were differentially expressed with the control [69,70]. According to a comparison of the epidermal bladder cell (EBC)-specific transcriptomes of the control and salt-treated *Mesembryanthemum crystallinum* L., the vesicular transport-related genes in the ER and Golgi were upregulated by salt. Furthermore, the H^+^-ATPase located in Golgi was induced at a 2.1-fold higher expression in the salt-treated plants [71]. The vesicle docking gene was upregulated by salt treatment in euhalophyte *Suaeda salsa*, which is possibly involved in compartmentalizing high concentrations of salt into vacuoles of leaf and young stem cells, leading to the succulence [72]. These data suggested at the genetic level that vesicle transport plays an important role in the process of salt secretion from the salt glands.

In recent years, the application of transcriptomic and genomic technology in the study of the recretohalophytes has led to the revelation of the candidate genes involved in salt secretion from the plant salt glands. Table 1 lists the candidate genes believed to be involved in salt secretion.

Some of the transcriptomes and proteomes of the recretohalophytes were constructed from plant leaves [82,83,84,85,86,87], which largely comprise mesophyll and epidermal cells. The salt glands represent only a small part of these tissues, meaning the differential expression of genes in these cells when undergoing salt secretion cannot typically be detected. It is therefore urgent to perform sequencing of the salt gland alone. Separating the salt glands by the enzymatic hydrolysis of the cell wall may be feasible [40], but no one has achieved sequencing analysis of the isolated salt glands.

## 6. Mechanism of Vesicular Transport in Salt Secretion

As early as 1999, when MacRobbie studied the transfer kinetics of tracer chloride, it was proposed that salt-filled vesicles were transferred from the cytoplasm to the vacuole in *Arabidopsis thaliana* [88]. According to Hamaji et al. (2009), when *Arabidopsis thaliana* is treated with high salt, vesicle movement is active; Na^+^ ions not only accumulate in the main vacuole, but also in the small compartments around the vacuole [89]. Flowers et al. (2019) presumed that there is another type of ion transport besides ion transporters in the halophytes, especially in the recretohalophytes [7]. Although plant models such as rice (*Oryza sativa*) and Arabidopsis (*Arabidopsis thaliana*) lack salt glands, they still have homologous cell structures and the orthologous gene families, which may be crucial effectors for sensing, transporting, and sequestering salt in the recretohalophytes, and thus require further study [18].

Under salt stress conditions, speed is of the essence, so transport proteins and proton pumps in the tonoplast or plasma membrane work in coordination to compartmentalize ions in the vacuole, export them from the cell, or transfer them out of the body via the vesicles [7]. The typical route of vesicle secretion is that the secreted substances are biosynthesized in the ER and then fuse with the plasma membrane or vacuole through the Golgi and TGN [79,90]. Several studies in *Arabidopsis thaliana* have reported that plant TGN-mediated trafficking responded to different types of biotic and abiotic stress. Plants with mutations in the TGN-specific marker genes (*SYNTAXIN OF PLANTS 61* (*SYP61*), *RAB GTPases A Group 2A* (*RABA2A*), *V-ATPase subunit VHA-a* (*VHA-A1*), or *SYNTAXIN OF PLANTS 43* (*SYP43*)) are very sensitive to salt stress [80,91,92,93]. Both the Arabidopsis *osm1* (*osmotic stress-sensitive mutant*) and *tno1* (*tgn-localization syp41-interaction protein*) mutants have aberrant localization of SYP61, and are very sensitive to salt and osmotic stress [91,92]. The AtVAMP7C family are v-SNAREs (vesicle soluble N-ethylmaleimide-sensitive factor attachment protein receptors) that function when the vesicle fuses with the tonoplast [77,94]. Mutants of this family lack the ability to fuse H_2_O_2_-containing vesicles with the tonoplast, thereby resulting in the retention of these vesicles in the cytoplasm [95]. Rab GTPases are pivotal regulators of membrane trafficking, recruiting selectively tethering or other specific-effector proteins in vesicular transport, tethering, and endosomal maturation [31]. Arabidopsis plants that lacked the four main RABA1 members exhibited hypersensitivity to salt stress [76]. The *Oryza sativa Rab11* (*OsRab11*) salt-induced gene, overexpressed *OsRab11* in Arabidopsis, exhibited enhanced tolerance to high salt stress and exogenous abscisic acid (ABA) compared to wild plants [96,97]. Salt stress (defined as 150 mM NaCl for 12 days in the soil) in chickpea (*Cicer arietinum* L.) results in a total CaRAB GTPases C group gene expression level that was 2–3 times higher than that of the control plants [98]. In addition, *CaRabA2* expression at 15 days was in correlation with (R^2^ = 0.905) with Na^+^ accumulation in chickpea (*Cicer arietinum* L.) leaves [99].

The pH balance along the endomembrane system is very important for cells. The luminal pH is rigorously regulated by the concerted function of V-ATPase (vacuolar -H^+−^ATPase), the Na/H^+^ anti-transporter, and chloride channels and anion transporters (CLCd) [100,101,102,103], and its disruption results in defects in secretion and stress tolerance. The *vha-a1* mutant, which lacks a TGN-located H^+^-pump is very sensitive to salt; however, the *vha-a2 vha-a3* double mutant lacking two vacuole-localized H^+^ pumps did not have an altered salt tolerance but instead absorbed more nitrate, which again illustrates the effect of the pH in the TGN lumen on salt stress [80,104]. The Arabidopsis TGN intraluminal pH is 6.1, while in the cytoplasm it is 7.5 [105], meaning more Na^+^ can be driven to the vesicles by H^+^ gradient [103,106], and that Na^+^/H^+^ antiporters use the H^+^ gradient to couple the passive transport of H^+^ to the transport of Na^+^ against their gradient. In Arabidopsis, NHX5 and NHX6 are located in the Golgi apparatus, TGN, and active intracytoplasmic vesicles, and are particularly sensitive to the vesicular transport inhibitor BFA. The growth of the *nhx5 nhx6* knockout mutant is inhibited, the cells are relatively small, and the salt sensitivity is increased [101,103,107]. The overexpression of *NHX5* in *Broussonetia papyrifera* significantly improved drought tolerance and salt tolerance of the transgenic plants [108]. Thus, NHX5 and NHX6 can transfer any excess salt in the cytoplasm to the vesicles during the vesicular transportation process.

Under salt stress, the increase of Na^+^ in plant cells will be accompanied by the accumulation of Cl^−^ ions, which can also cause cytotoxicity due to the disturbance of the electric charge. The Cl^−^ channels CLCd and CLCf are localized in the Golgi apparatus and the TGN. These channels co-localize with VHA-a1 and adjust the pH balance in the TGN lumen through the transport of Cl^−^ and NO_3_^−^ [78,79,109]. The Arabidopsis T-DNA insertion mutant *clcd-1* is more sensitive to concanamycin A, a V-ATPase inhibitor, than the wild type [78]. Lu et al. (2020) downregulated the TGN-located SNARE gene *LbSYP61* in *Limonium bicolor* using virus-induced gene silencing, leading to a significant decrease in the secretory ability of the salt glands [67]. Exogenous melatonin enhances salt secretion of *L. bicolor* leaves by upregulating the expression of ion transporter and vesicle transport genes [110]. Drakakaki et al. (2012) performed a proteomic analysis of SYP61-labeled vesicles, revealing that they contained many SYP61, SYP121, and SNARE complexes, cellulose synthase complexes (CSCs), and ECHIDNA (ECH) proteins, which indicates that SYP61 plays important roles in extracellular transport [75,111]. According to Hachez et al. (2014), SYP61 and SYP121 can cooperatively regulate the transport of PIP2;7 (plasma membrane intrinsic proteins 2;7) from the Golgi to the plasma membrane [112]. A salt treatment resulted in the inhibition of the *PIP2;7* promoter activity and a significant reduction in *PIP2;7* mRNA abundance [113]. Furthermore, a subpopulation of cortically tethered small CesA compartments was found to be colocalized with SYP61 [111,114]. These data support the hypothesis that some vesicular transport genes, such as *SYP61*, participate in plant salt secretion.

The responses of cells to abiotic and biotic stress depend on the regulation of vesicle trafficking, which ensures the correct positioning of specialized proteins in sensing stress stimuli and responding accordingly. Eukaryotes embed hydrophobic cargo within the lipid bilayer of the vesicle, while soluble cargo is contained within the vesicle, ensuring the exchange of large molecules and proteins between different compartments and membrane systems [33,90]. Under salt stress, we propose that excessive ions in the cytoplasm are pumped into the vesicle by NHX5/6 and the CLCd anti-transporter, maintaining the intracytoplasmic ion balance. The osmotic potential and pH balance of the vesicles are adjusted by the aquaporin PIP2;7 and the proton pump VHA-a1. The vesicles containing salt move to the plasma membrane, fuse with the membrane, and finally release the salt into the extracellular spaces, such as the cell wall or out of the plant body (Figure 2).

## 7. Conclusions and Perspectives

Salt glands are unique epidermal structures of the recretohalophytes, which undertake the active physiological process of salt secretion [3,28,45,115]. In this review, we discussed recent research progress that showed vesicular trafficking is important for salt secretion, and revealed the underlying mechanisms. This review updates and supplements the salt gland secretion model; however, the detailed mechanisms by which excess ions are secreted and homeostasis is maintained are complicated, lack direct evidence, and therefore require further study. The lack of genomic and transcriptomic information for isolated salt glands means the mechanisms of salt secretion are still unclear. How the recretohalophytes integrate a large number of environmental signals to finely regulate their endomembrane trafficking is yet to be fully elucidated. Future research should focus on the molecular mechanisms by which ions are transported from the mesophyll cells into the salt glands and out of these secretory cells.

The publication of additional annotated genomes and the construction of an efficient transform system for a recretohalophyte species will greatly enhance their study. We hope to complete the genome sequencing of halophytes with salt glands as soon as possible, and screen out the recretohalophyte mutants that can enhance or weaken the secretion of salt or even loss of the salt secretion, so as to identify the key genes functioning in the salt glands. Tan et al. (2010) made it possible to isolate salt glands without mesophyll cells for single-cell sequencing, from which the unique genes of the salt glands or key genes for salt secretion could be identified [40]. The application of genome-editing technology such as CRISPR-Cas9 and VIGs could be used to verify the function of the key genes for salt secretion, such as the transformation system of *L. bicolor* has established [116]. We believe that the mystery of salt gland secretion will be solved in the near future.

## Figures and Tables

**Figure 1 ijms-22-02203-f001:**
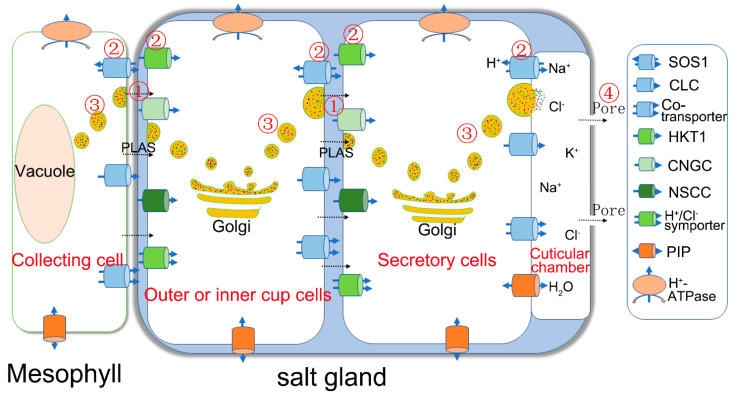
Simplified illustration of salt secretion from plant salt glands. Na^+^ is transported into the salt gland from the collecting cell, which covers the salt gland, and consists of a huge vacuole and a shrinking cytoplasm, through the plasmodesmata (pathway 1), membrane-bound transporters (pathway 2) such as SOS1 in the transfusion zone, or via vesicular transport (pathway 3). In the salt gland (blue), the ions can be directly transported into the intercellular space of the outer or inner cup cells and the secretory cells via the different pathways (①, ②, ③). The ions are parceled into vesicles for transport from the cytosol to the plasma membrane, then secreted out of the salt gland cells via pathways ② and ③. The ions are eventually forced out of the secretory pores ④ at the top of the salt gland as result of the high hydrostatic pressure. Ion transporters in charge of influx (green) and efflux (blue) are asymmetrically distributed in the plasma membrane of salt gland cells. PLAS, plasmodesmata; HKT1, high-affinity K^+^ transporter 1; CNGC, cyclic nucleotide-gated cation channel; NSCC, non-selective cationic channel; PIP, plasma membrane intrinsic protein; NHX, Na^+^/H^+^ antiporter; SOS1, Na^+^/H^+^ antiporter; CLC, chloride channel.

**Figure 2 ijms-22-02203-f002:**
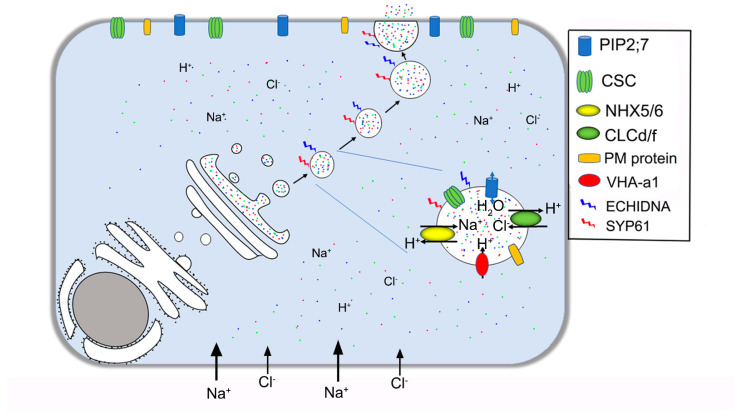
Vesicular transport in the salt gland secretion process. Excessive ions in the cytoplasm are pumped into the SYP61- or ECH-labeled vesicles by NHX5/6 and the CLCd/f anti-transporter. The osmotic potential and pH balance of the vesicles are adjusted by the aquaporin PIP 2;7 and the proton pump VHA-a1. SYP61, Syntaxin of plants 61; ECH, ECHIDNA protein; NHX5/6, Na^+^/H^+^ antiporter 5/6; PIP2;7, plasma membrane intrinsic protein 2;7; CLCd/f, chloride channel d/f.

**Table 1 ijms-22-02203-t001:** Candidate genes involved in salt secretion in the recretohalophytes.

Genes	Function	Location	Reference
*ECHIDNA*	Vesicle formation	TGN/EE	Gendre et al. 2011 [73]; Gendre et al. 2013 [74]
*SYP61*	Membrane fusion	TGN/EE, PM	Drakakaki et al. 2012 [75]; Lu et al. 2020 [67]
*RABA1A*	Regulation of vesicle trafficking	TGN/EE	Asaoka et al. 2013 [76]
*VAMP7C*	Vesicle transport	TGN/EE	Leshem et al. 2010 [77]
*NHX5/6*	Exchange H^+^/Na^+^ or K^+^	TGN/EE, Golgi	Asaoka et al. 2013 [76]
*CLCd*	Exchange H^+^/Cl^−^ or NO_3_^−^	TGN/EE, Golgi	Fecht–Bartenbach et al. 2007 [78]; Heard et al. 2015 [79]
*VHA-a1*	H+-ATPase	TGN/EE	Krebs et al. 2010 [80]; Zhou et al. 2016 [81]

PM: plasma membrane; TGN/EE: trans-Golgi network/early endosomes.

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
