# Peer review of "Current Understanding of Role of Vesicular Transport in Salt Secretion by Salt Glands in Recretohalophytes"

_ijms, 2021, doi:10.3390/ijms22042203_

Round 1

Reviewer 1 Report

The submitted review article by Lu C et al. summerizes that mechanism of salt secretion by vesiclular transport in recretohalophytes.

The manuscript is well written, and I enjoyed reading it and knew current progress of the relevant studies.

What I wanted to know the mechanism how Cl- is selectively stored and maintained inside vesicle was clearly explained in the text and with the figure 2.

I have small issue for the revision as follows,

The components shown in the illustration of Figure 1 shouldn't be overlapped with the characters, e.g., the symbols of channels.

Both in Fig1 and Fig2, the fonts, such as "Cl-", "K+", and "Na+" should be larger, too.

Author Response

Point 1: What I wanted to know the mechanism how Cl- is selectively stored and maintained inside vesicle was clearly explained in the text and with the figure 2.

Response: According to Fecht-Bartenbach et al. (2007) and Shen J et al. (2013), Cl- may be stored selectively stored and maintained inside vesicle by antiporter CLCd or CLCf.

Point 2: I have small issue for the revision as follows,

The components shown in the illustration of Figure 1 shouldn't be overlapped with the characters, e.g., the symbols of channels.

Both in Fig1 and Fig2, the fonts, such as "Cl-", "K+", and "Na+" should be larger, too.

Response: Done. We have re-edited the figures in the article and increased the figures resolution.

Reviewer 2 Report

The work “Current understanding of role of vesicular transport in salt secretion by salt glands in recretohalophytes” by Lu et al. consists of a Review paper that reviews the relevant literature—articles and research on the proposed topics.

Integrating the current knowledge on vesicular traffic and salt secretion is no easy task since the two bodies of knowledge rarely intersect.

The authors very elegantly and intelligently build their line of reasoning with a balanced analysis of the available information, which they synthesize, and interpret in a meaningful way, achieving the difficult task of allowing readers to make sense of all available information.

They integrate seamlessly morphological and structural features with cytological and physiological knowledge, bringing in also Insights from transcriptomic and genomic sequencing.

Several lines of research and hypotheses that explain the salt secretion process of plant salt glands are presented in a structured and a rational way leading to the authors building  a proposed, illustrated model of salt secretion from plant salt glands.

They pinpoint the growing evidence in favour of vesicular transport in plants and evidence in favour of the movement of ions in vesicles.

In this way, this review qualifies as a systematic review rather than a narrative one since the authors critically attempt to build a hypothesis based on published evidence.

The review addresses an important aspect of the field and tackles a well-defined issue so that it may interest readers of various fields, such as those involved in intracellular vesicular traffic, conventional and unconventional routes and how this impacts and interplays in abiotic stress situations like salt stress and draught.

Maybe the slightly weaker section pertains transcriptomics and expression profiles of intracellular vesicle trafficking genes genes and salt accumulation.

Also there is not a strong emphasis on the goal of the review to identify research questions.

Although the comprehensiveness of the literature search is quite good as is the state of the art, two recent articles that are not mentioned in the review but could be important:

1 - Sweetman, C., Khassanova, G., Miller, T.K. et al. Salt-induced expression of intracellular vesicle trafficking genes, CaRab-GTP, and their association with Na+ accumulation in leaves of chickpea (Cicer arietinum L.). BMC Plant Biol 20, 183 (2020). https://doi.org/10.1186/s12870-020-02331-5.

2 - Zhang, X., Yao, Y., Li, X. et al. Transcriptomic analysis identifies novel genes and pathways for salt stress responses in Suaeda salsa leaves. Sci Rep 10, 4236 (2020). https://doi.org/10.1038/s41598-020-61204-x

Author Response

Point 1: Sweetman, C., Khassanova, G., Miller, T.K. et al. Salt-induced expression of intracellular vesicle trafficking genes, CaRab-GTP, and their association with Na+ accumulation in leaves of chickpea (Cicer arietinum L.). BMC Plant Biol 20, 183 (2020).  87https://doi.org/10.1186/s12870-020-02331-5.

Response: Thanks for the suggestions. We read and cited this study. The details are provided at page 7 line 284-285.

Point 2 : Zhang, X., Yao, Y., Li, X. et al. Transcriptomic analysis identifies novel genes and pathways for salt stress responses in Suaeda salsa leaves. Sci Rep 10, 4236 (2020). https://doi.org/10.1038/s41598-020-61204-x

Response: Thanks for the suggestions. The details are provided in lines 231-233 at page 6. The article is added to the reference (72).

Reviewer 3 Report

The Review entitled "Current understanding of role of vesicular transport in salt secretion by salt glands in recretohalophytes" might deserve publication after some revisions and suggestions as reported below:

Lines 56-57: add references.

Lines 69-73: add references.

Line 77: add species. Is “Spartina foliosa”?

Figure 1: is it original or does it need bibliographic references? I would make the transporters a little bigger because the writings are badly read and go on the arrows. Or I would enter a legend with different colors as in figure 2. Also, I would like to know what they represent the numbers in brackets in the caption, the references or the pathways?

Lines 114-119: add references.

Line 142: after "Avicennia" remove the point.

Line 211: "Insights from transcriptomic and genomic sequencing". Since we are talking about omics applications, I would insert it in the title, otherwise the title does not indicate the presence of proteomic studies.

Line 245: "Mechanism". Explain better in the title of the paragraph which mechanisms it is reported.

Line 252: add the species to Arabidopsis.

Lines 248-265: the authors very often cite studies on Arabidopsis and Oryza sativa, I was wondering if such studies were also carried out in halophytes? If necessary, supplement with studies on recretohalophytes.

In paragraph 5, insert the species subject to these studies.

Finally, I would like to know if you also know the size of these glands and vesicles?

One last suggestions for authors I would advise you to check if the most recent bibliographical references have been included.

Author Response

Reviewer #3: The Review entitled "Current understanding of role of vesicular transport in salt secretion by salt glands in recretohalophytes" might deserve publication after some revisions and suggestions as reported below:

Point 1Lines 56-57: add references.

Response: Done in lines 57-58 at page 2 and marked in red in the manuscript.

Point 2Lines 69-73: add references.

Response: Done in line 73 at page 2 and marked in red in the manuscript.

Point 3Line 77: add species. Is “Spartina foliosa”?

Response: Thanks for the suggestions, species is Spartina anglica, added in line 77 at page 2 and marked in red in the manuscript.

Point 4Figure 1: is it original or does it need bibliographic references? I would make the transporters a little bigger because the writings are badly read and go on the arrows. Or I would enter a legend with different colors as in figure 2. Also, I would like to know what they represent the numbers in brackets in the caption, the references or the pathways?

Response: Done. We have re-edited the figures in the article and increased the figures resolution. The numbers in figure 1 are the pathways.

Point 5Lines 114-119: add references.

Response: Done in line 115,116 at page 3, line 126 at page 4 and marked in red in the manuscript.

Point 6Line 142: after "Avicennia" remove the point.

Response: Done in line 145 at page 4.

Point 7Line 211: "Insights from transcriptomic and genomic sequencing". Since we are talking about omics applications, I would insert it in the title, otherwise the title does not indicate the presence of proteomic studies.

Response: Thanks for the suggestions. We have accepted your advice and changed the title into " Insights from omics applications " in line 213 at page 5.

Point 8Line 245: "Mechanism". Explain better in the title of the paragraph which mechanisms it is reported.

Response: Thanks for the suggestions. We agree with you and changed the title into " Mechanism of vesicular transport in salt secretion " in line 250 at page 6.

Point 9Line 252: add the species to Arabidopsis.

Response: Done in lines 262-263 at page 6.

Point 10Lines 248-265: the authors very often cite studies on Arabidopsis and Oryza sativa, I was wondering if such studies were also carried out in halophytes? If necessary, supplement with studies on recretohalophytes.

Response: There are few studies on recretohalophytes in mechanism of secretion. In recent years, the studies focus on the development of salt gland. Although model plants such as rice and Arabidopsis lack salt glands, they still have similar protein sorting, trafficking and targeting and the orthologous gene families, which may be crucial effectors for sensing, transporting, and sequestering salt in the recretohalophytes, and thus require further study.

Point 11In paragraph 5, insert the species subject to these studies.

Response: Thanks for the suggestions. We have accepted your advice and inserted the species. Marked in red in the manuscript.

Point 12Finally, I would like to know if you also know the size of these glands and vesicles?

Response: Different recretohalophyte species have different sizes of salt glands, and salt gland consist of 2-40 cells. Some studiers summarized the sizes of vesicles in plants, where clathrin-independent vesicles are 30-100 nm or more in diameter.

Point 13One last suggestions for authors I would advise you to check if the most recent bibliographical references have been included.

Response: Thanks for the suggestions. We have accepted your advice and consulted recent reference.

Round 2

Reviewer 3 Report

The authors answered in an exhaustive manner to all suggestions proposed. One last suggestion concerns the formatting of reference numbers throughout the text: replace the comma with the - or vice versa, for example [3-18] line 57. The comma is missing at reference 72 after Zhang.

Following these minor suggestions, in my opinion the review has an excellent scientific value and deserves to be published